# Fluorescence Line Height Extraction Algorithm for the Geostationary Ocean Color Imager

**Min Zhao** [1,2]**, Yan Bai** [1,2,*]**, Hao Li** [2]**, Xianqiang He** [1,2,3]**, Fang Gong** [2] **and Teng Li** [2]

1 School of Oceanography, Shanghai Jiao Tong University, Shanghai 200030, China; zhaomin@sjtu.edu.cn (M.Z.); hexianqiang@sio.org.cn (X.H.)
2 State Key Laboratory of Satellite Ocean Environment Dynamics, Second Institute of Oceanography, Ministry of Natural Resources, Hangzhou 310012, China; lihao@sio.org.cn (H.L.); gongfang@sio.org.cn (F.G.); liteng@sio.org.cn (T.L.)
3 Southern Marine Science and Engineering Guangdong Laboratory (Guangzhou), Guangzhou 511458, China
* Correspondence: baiyan@sio.org.cn

**Abstract:** Chlorophyll fluorescence is an important indicator of the physiological state of phytoplankton in water bodies. The new generation of ocean color satellite remote sensors usually sets fluorescence bands to detect the phytoplankton fluorescence line height (FLH). Yet, the Geostationary Ocean Color Imager (GOCI) offers no FLH products so far, and the FLH results calculated using the fluorescence band (680 nm) and the two baseline bands (660 and 745 nm) have numerous negative values and are quite different from the FLH products of other satellite ocean color sensors. To address this problem, we established an FLH retrieval algorithm suitable for GOCI. We simulated the spectral datasets of different water types using the radiative transfer model HydroLight and established the band conversion relationship from 680 to 685 nm based on the simulated datasets. The remote sensing reflectance after band conversion was applied to the FLH product inversion, significantly reducing the number of negative FLH values and appreciably improving data availability for GOCI FLH products (from 14.78% to 66.73% on average). The new FLH product has a good correlation with the field-measured data ($R^2 = 0.73$), and the relative error was 6.95%. Moreover, after band conversion, the FLH products retrieved by GOCI are in good agreement with the FLH products of MODIS, and fusion products can be further produced to improve the spatiotemporal resolution of the data. In addition, the radiative transfer simulation datasets also revealed that changes in solar zenith angle have little effect on FLH inversion. The hourly GOCI-derived FLH has good spatiotemporal continuity and can clearly reflect the diurnal variation of FLH. It can provide a stable FLH algorithm for further recovery of time-series GOCI FLH products and research on diurnal changes in FLH.

**Keywords:** ocean color; GOCI; fluorescence line height; chlorophyll; diurnal change

## 1. Introduction

Chlorophyll fluorescence is an important energy release process after phytoplankton absorb sunlight. It prevents chloroplasts from absorbing light energy other than through photosynthesis and minimizes the loss of strong light burns. It is also closely related to the biomass and photosynthesis process of phytoplankton. Through water-leaving radiance or remote sensing reflectance (Rrs), the chlorophyll fluorescence signal excited by sunlight can be extracted to analyze the physiological changes of phytoplankton [1,2]. A commonly used parameter is the fluorescence line height (FLH). By relying on the fluorescence signal excited by the sun, FLH can be used to estimate the degree to which the radiance produced by chlorophyll fluorescence in water is higher than that of pure water [3], and FLH is one of the main indicators used to analyze the physiological changes of phytoplankton [4]. FLH relies on three bands for calculation: the central wavelength of the maximum value of chlorophyll fluorescence (near 685 nm) and the two other bands used to generate the baseline under the fluorescence peak, which are located on opposite sides of the fluorescence peak [5].

Nowadays, many ocean color sensors have fluorescent bands (e.g., the Moderate Resolution Imaging Spectroradiometer (MODIS) and the Ocean and Land Color Instrument (OLCI) aboard the polar-orbiting Aqua and Sentinel-3A/Sentinel-3B satellites, respectively, and the Geostationary Ocean Color Imager (GOCI) aboard the Communication, Ocean and Meteorological Satellite (COMS), the first stationary ocean color satellite). MODIS has three FLH calculation bands, which are 667, 678, and 748 nm, and NASA officially provides FLH products. The OLCI has multiple FLH calculation bands (e.g., 665, 674, 681, 710, and 761 nm).

Many scientists have studied fluorescence line height retrieval algorithms for different satellite sensors. Morel and Prieur [6] first pointed out that there is a strong positive correlation between the fluorescence peak height and chlorophyll concentration, and they proposed the idea of using the FLH to invert the concentration of chlorophyll a (Chla). With the continuous launch of ocean color satellites, remote sensing on FLH has also been continuously developing. After considering the influence of the atmosphere on fluorescence, Gower et al. [7] provided two optimal band combinations (one set of bands at 659.5–672.5, 673.5–688.5, and 746.2–757.8 nm and the other at 659.5–672.5, 673.5–688.5, and 708.8–713.9 nm) for the fluorescence line height measurement by satellite; these enable effective avoidance of the influence of the absorption of water vapor and oxygen in the atmosphere on the FLH retrieval. Letelier et al. [8] analyzed the feasibility of using MODIS to detect the chlorophyll fluorescence signal, and their results showed that effective FLH products can be produced by using the three bands (667, 678, and 748 nm) of MODIS. Fiorani et al. [9] evaluated the accuracy of the chlorophyll concentration products retrieved from the FLH products of MODIS using the measured data of the Southern Ocean voyage from 2002 to 2003. Their results showed that the accuracy of the chlorophyll concentration products was better than that of the commonly used blue–green band ratio algorithm. Hoge et al. [10] used an airborne laser to induce chlorophyll fluorescence signals in the coastal waters of the Middle Atlantic Bight in the western North Atlantic, successfully verifying the accuracy of the FLH products of MODIS in different water bodies. The correlation coefficient between the airborne laser data and the MODIS FLH products was 0.85. Gower and King [11] verified the accuracy of MERIS satellite chlorophyll fluorescence products using cruise data in the coastal waters of western Canada, and their results revealed that the chlorophyll concentration accuracy obtained by the FLH was comparable to that obtained using the blue–green band ratio method. In addition to retrieving chlorophyll concentration, fluorescence height information can also be used to identify red tide and phytoplankton fluorescence efficiency. For example, Son et al. [12] used the fluorescence products of MODIS to distinguish red tide water bodies (especially *Cochlodinium polykrikoides* red tide) from non-red tide water bodies. Behrenfeld et al. [13] studied the response of phytoplankton fluorescence characteristics to changes in nutrient concentration. By comparing the fluorescence efficiency and iron distribution, they found that the two were closely related, and the region with lower iron content had higher fluorescence efficiency. Robert et al. [14] studied the response of phytoplankton fluorescence characteristics to changes in light intensity and found that there is a strong correlation between the instantaneous photosynthetically active radiation and the ratio of phytoplankton FLH to chlorophyll concentrations. This finding could provide the basis for further research on nonphotochemical quenching phenomena.

The geostationary satellite COMS was deployed in 2010. Its GOCI instrument can acquire data in eight spectral bands with a spatial resolution of 500 m and a revisit time of ~1 h, which enables the real-time monitoring of marine ecological environments from space [15,16]. Although GOCI has not officially provided FLH products so far, the conditions for calculating FLH are present. It has the chlorophyll fluorescence peak band of 680 nm, and there are baseline bands of 660 and 745 nm on opposite sides of the fluorescence peak. Therefore, theoretically, an FLH product can be calculated by using GOCI's data. Son et al. [17] distinguished *Cochlodinium polykrikoides* red tide from non-red tide water bodies based on the chlorophyll fluorescence signal of GOCI. Kim et al. [18] evaluated the accuracy of GOCI's chlorophyll products processed using different

algorithms by inputting measured data from multiple cruises around the Korean Peninsula from 2010 to 2014, in which the FLH algorithm for GOCI was computed as $FLH = Rrs(680) - \left[ Rrs(660) + (Rrs(745) - Rrs(660)) \times \frac{(680-660)}{(745-660)} \right]$ (where Rrs(660), Rrs(680), and Rrs(745) are the remote sensing reflectances at 660, 680, and 745 nm, respectively). The algorithm using FLH to invert the chlorophyll concentration (Chla = $c_0 FLH^{c_1}$, where $c_0$ and $c_1$ are empirical coefficients) has the same accuracy as the OC3 algorithm (with an absolute percentage deviation (APD) of ~40%) in water with a chlorophyll concentration of <1 mg/m$^3$. However, when the chlorophyll concentration is 1–3 mg/m$^3$, the relative error exceeds 70%, and when the chlorophyll concentration exceeds 3 mg/m$^3$, the accuracy drops rapidly, and the APD even exceeds 500%. However, we compared the FLH calculated from GOCI data at 660, 680, and 745 nm with the MODIS FLH product results (the same as from Kim et al.'s algorithm), as shown in Figure 1 on 18 October 2011, 27 April 2012, and 7 December 2013, at 1:30 pm. From the GOCI and MODIS observation data, it can be clearly seen that, in different regions and in different seasons, there are obvious differences in the FLH results calculated by using the two ocean color sensors. The effective value coverage of the FLH products of MODIS is much higher than that of GOCI; and the FLH inversion results of GOCI have numerous spatial discontinuities. Even under clear sky conditions, the FLH inversion results from GOCI exhibit numerous negative values. In general, effective FLH products cannot be achieved using the fluorescence bands of GOCI. Therefore, constructing an FLH algorithm suitable for GOCI is needed.

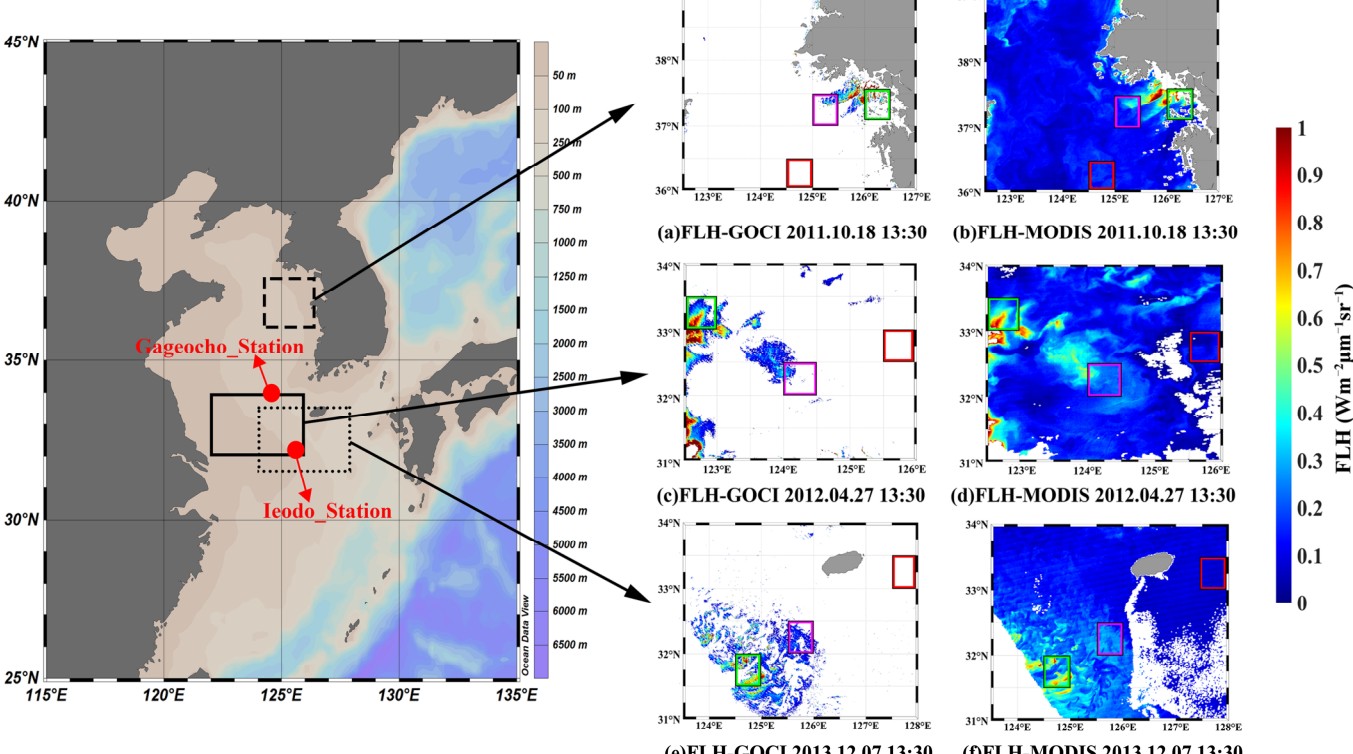

**Figure 1.** FLH inversion results of GOCI compared with the FLH product of MODIS. The white area in the figure is the invalid value area, which may be caused by cloud cover or by failure of the FLH algorithm. The framed areas in panels (**b**,**d**,**f**) are the areas selected for study. The green frame is the nearshore water area, the purple frame is the continental shelf water area, and the red frame is the clear ocean water area. The two red dots are the locations of the AERONET-OC stations used in this study (The station names are: Gageocho_Station and Ieodo_Station).

In this study, an FLH inversion model suitable for the GOCI observation band was developed. First, the radiative transfer model HydroLight was used to simulate various

water body conditions, and the problems caused by using the original GOCI band to calculate FLH were analyzed. Then, a new FLH algorithm for GOCI was established based on the fluorescence characteristics of the simulated dataset. The accuracy of the established algorithm was verified with the in situ data from two cruises conducted in the East China Sea from 2 to 13 August 2016 and from 16 to 26 August 2016, and the cross-comparison with the FLH product of another satellite, MODIS, shows the reliability of the algorithm. Finally, the established algorithm was used to process hourly GOCI satellite data near the Korean Peninsula to reveal the hourly variation of FLH.

## 2. Materials and Methods

### 2.1. Satellite Data

We downloaded GOCI data from the Korea Ocean Satellite Center website (http: //kosc.kiost.ac.kr/ (accessed on 15 February 2022)). GOCI observes once an hour from 8:30 to 15:30 (local time of the location of the observation center), with 36°N and 130°E as the observation center, covering an area of ~2500 km × 2500 km. It has high temporal resolution (1 h) with a spatial resolution of 500 m, which makes it possible to monitor the diurnal change of ocean and inland waters. Because the level-2 data of GOCI do not include the FLH product and near-infrared-band Rrs products, we downloaded the level-1 GOCI data, then processed the data with SeaDAS 8.1 using a near-infrared atmospheric correction algorithm [19] and generated the Rrs products of all spectral bands.

In addition to the GOCI data, we also obtained the MODIS data. MODIS is one on the key instruments aboard the Aqua satellite. In this study, MODIS data were used to cross-compare with the FLH products of GOCI. MODIS level-1 and level-2 data were downloaded from NASA's ocean color website (https://oceancolor.gsfc.nasa.gov/data/aqua/ (accessed on 15 February 2022)). The observation time of MODIS is around 13:30 local time, and its spatial resolution of the ocean color band is 1 km. The FLH products of MODIS used in this study come from the downloaded level-2 data by using the 667, 678, and 748 nm bands. The MODIS visible-band Rrs products were obtained from the downloaded level-1 data processed by SeaDAS 8.1 using the standard near-infrared atmospheric correction algorithm.

### 2.2. In Situ Data

To evaluate the accuracy of the established FLH algorithm, we used the in situ data from two cruises conducted in the East China Sea. The in situ measurement dates were from 2 to 13 August 2016 and from 16 to 26 August 2016. There are a total of 117 stations included in the two cruises, as shown in Figure 2a. Some stations had no spectral data because of weather conditions, and this left a total of 77 stations with water spectral data (Figure 2b). By using the water spectral data at 660, 685, and 745 nm, the in situ measured FLH values were obtained. It can be seen that the fluorescence signal is basically high near the shore and low farther from the shore.

During the cruises, a portable spectrometer (ASD, Inc., Atlanta, GA, USA), having a spectral range of 350–2500 nm and a spectral resolution of 1 nm, was used to measure the Rrs of the water body. The measurement was performed in accordance with the NASA Ocean Optics Specification [20]. On this basis, we calculated Rrs as follows:

$$\text{Rrs} = \frac{r(L_t - L_s \times \rho)}{\pi L_r},\tag{1}$$

where $L_t$ is the upward radiance above the water surface, $L_s$ is the downward radiance of the sky, $r$ is the reflectance of the reference gray plate, $L_r$ is the upward radiance from the reference gray plate, and $\rho$ is the reflectance of the water surface, which is set as 0.028 [21]. After the Rrs data were calculated, the FLH value was calculated using the fluorescence line height formula, which was used for the subsequent algorithm accuracy test.

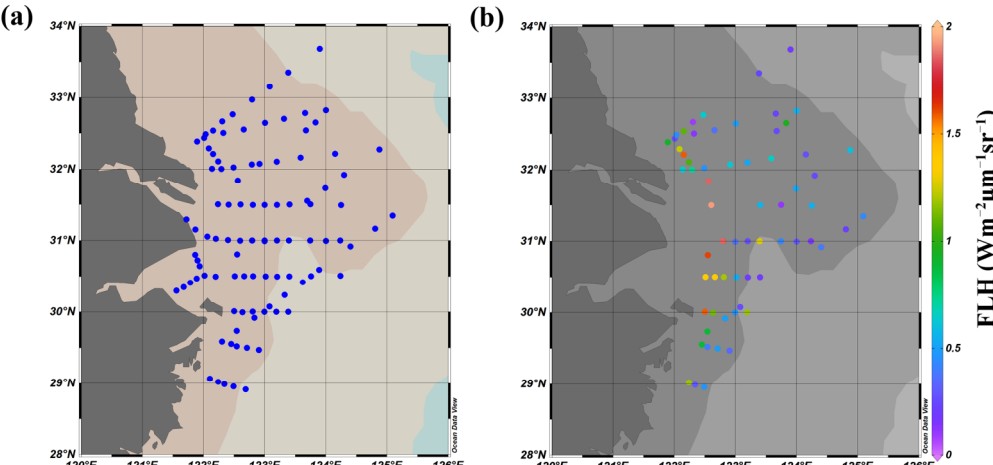

**Figure 2.** Sampling stations of two cruises in the East China Sea in 2016. (**a**) All measurement stations. (**b**) Stations with effective FLH data.

To enhance the algorithm validation, we added in situ Chla data derived by AERONET-OC. AERONET-OC was developed to support atmospheric research at different scales with measurements taken from CE-318 solar photometers installed on offshore platforms such as lighthouses, ocean monitoring towers, and oil towers. The Rrs values measured by the AERONET-OC stations have played an important role in the verification of satellite ocean color data. However, because there is no near-infrared band, the AERONET-OC site does not provide FLH products. Here, we use the Chla obtained from AERONET-OC dataset to calculate the FLH and then to verify the accuracy of the algorithm. The algorithm of Chla being invert FLH was from Kim et al. [18] mentioned above. The AERONET-OC dataset used contains 138 points of Chla data during 2011 to 2018 in our study area, covering different seasons. The locations of the stations selected in this study are shown in Figure 1.

### 2.3. Fluorescence Signal Calculation

The general fluorescence line height calculation relies on three wavelengths: the maximum value of chlorophyll fluorescence, which is the fluorescence peak (near 685 nm), and the two other bands located on opposite sides of the fluorescence peak [22]. The calculation formula is as follows:

$$\text{FLH} = \text{Rrs}_F - [\text{Rrs}_R + \frac{\lambda_R - \lambda_F}{\lambda_R - \lambda_L}(\text{Rrs}_L - \text{Rrs}_R)], \qquad (2)$$

where $\text{Rrs}_F$ is the remote sensing reflectance at the fluorescence peak band; $\text{Rrs}_R$ is the remote sensing reflectance at the shorter baseline band; $\text{Rrs}_L$ is the remote sensing reflectance at the longer baseline band; and $\lambda_F$, $\lambda_L$, and $\lambda_R$ are the central wavelengths of the fluorescence band and the two baseline bands, respectively.

### 2.4. Radiative Transfer Simulations

In this study, we used HydroLight 5.0 [23] (Sequoia Technology Company, Dumfries, VA, USA) to simulate water spectral under different water types. The constructed spectral dataset was then used to improve the current GOCI FLH algorithm as described in the following.

Li et al. [15] defined the type of water body when they studied the radiation sensitivity and signal detectability of ocean color satellite sensors at large solar zenith angles (SZAs). Based on Li's work, we defined the radiation distribution data in the water body (e.g., irradiance, reflectance, and diffuse attenuation function) and then set up the concentrations of various parameters, such as chromophoric dissolved organic matter (CDOM), total suspended matter (TSM), Chla, and surface and bottom boundary conditions. The radiation

distribution was calculated using HydroLight. We chose built-in models or defined inputs in FORTRAN subroutines based on the specific environmental conditions being simulated, and the absorption and scattering properties of pure water were adopted from Pope and Fry [24]. The Petzold phase function was adopted to define the scattering properties of particles, and other parameters were the default options [25,26]. A cloudless environment was selected for the sky condition, and infinite depth was selected for the bottom boundary condition. Finally, HydroLight calculates the radiance in water as a function of depth, direction, and wavelength by solving the scalar radiative transfer equation. The water-leaving radiance and Rrs are also obtained in the simulations. Values in Table 1 are the inputs for the HydroLight simulations, including the concentration of Chla, TSM, absorption coefficient of CDOM at the wavelength of 443 nm (aCDOM(443 nm)), and solar zenith angles in a certain step width. In water bodies with low turbidity, the simulation step width is relatively small, because slight changes in water color components will cause large changes in satellite spectral signals. However, for high-turbidity water, the satellite signal may be oversaturated, and the relative change is also small, so the simulation step width is relatively large. Taking into account the simulation efficiency and the coverage of water body types, we use the simulated concentrations as given in Table 1.

**Table 1.** Setup for the HydroLight simulations.

| Parameter | Values |
|---|---|
| Chla (mg/m$^3$) | 0.1; 0.2; 0.5; 1; 2; 5; 10; 20; 50; 100; 150; 300 |
| TSM (mg/L) | 0.1; 0.2; 0.5; 1; 2; 5; 10; 20; 50; 100; 150; 300 |
| aCDOM(443 nm) (m$^{-1}$) | 0.1; 0.2; 0.5; 1; 2; 5; 10 |
| SZA (°) | 0; 20; 40; 60; 70; 80 |

To study the migration of fluorescence bands in different water environments, we set up three specific water types: clear ocean water, continental shelf water, and nearshore water. Table 2 lists the specific concentrations of the three components in these three water types.

**Table 2.** Parameter settings for three different water types [19].

| Water Condition | Parameter | | |
|---|---|---|---|
| | Chla (mg/m$^3$) | TSM (mg/L) | aCDOM(443 nm) (m$^{-1}$) |
| Clear ocean water | 0.1 | 0.1 | 0 |
| Continental shelf water | 1 | 1 | 0.15 |
| Nearshore water | 5 | 1 | 0.2 |

*2.5. Evaluation of Performance*

For spatiotemporal matching of the in situ data and satellite data, we adopted the method of He et al. [27]. Specifically, a 3 × 3 pixel spatial window and ±3 h temporal window were used to match up the satellite and in situ data. In addition, the following conditions must be met:

- The percentages of valid pixels in the 3 × 3 box (excluding land pixels) were first checked. If the percentage was >50%, the box was adopted for the following quality assessment.
- The means and standard deviations (SDs) of valid values within the box were calculated. We discarded pixels with values outside the range of mean ± 1.5 SD.
- The means and SDs for the remaining valid pixels were recalculated and the coefficients of variation (CVs) were determined (where CV = SD/mean) to test spatial heterogeneity. If CV was <0.15, the box was adopted for matching the measured data with satellite data.

Performance assessment was based on statistical parameters, including root mean square deviation (RMSD), APD, and relative percentage deviation (RPD), defined as follows:

$$\text{RMSD} = \sqrt{\frac{\sum_{i=1}^{N}(Y_i - X_i)^2}{N}}, \quad (3)$$

$$\text{APD} = 100\% \times \frac{1}{N} \sum_{i=1}^{N} \frac{|Y_i - X_i|}{X_i}, \quad (4)$$

$$\text{RPD} = 100\% \times \frac{1}{N} \sum_{i=1}^{N} \frac{Y_i - X_i}{X_i}, \quad (5)$$

where $Y_i$, $X_i$, and $N$ represent the algorithm-retrieved values, in situ values, and sample number, respectively. RPD is a predictor of the systematic error or mean relative bias, and APD is the absolute accuracy of derived products relative to known data (in situ or reference data).

## 3. Establishment of the FLH Algorithm for GOCI

### 3.1. Analysis of GOCI and MODIS Chlorophyll Fluorescence Signals

Figure 1 shows the comparison between FLH derived by GOCI Rrs data and the FLH product of MODIS. Under clear-sky conditions, there are numerous negative values in the GOCI-derived FLH. To analyze the reasons for the negative values, we selected three different regions (Figure 1) in the East China Sea on 18 October 2011, 27 April 2012, and 7 December 2013, respectively, and analyzed the spectrum of the Rrs from GOCI.

Figure 3 shows the spectrum of average Rrs values in the three selected regions. It can be seen that, in nearshore water bodies and continental shelf water bodies with relatively high chlorophyll concentrations, the Rrs value at 680 nm is higher than the line connecting 660 and 745 nm as the baseline; therefore, the calculated FLH value is positive. However, in clear ocean water, the Rrs value at 680 nm is lower than the baseline connecting 660 and 745 nm; that is, the baseline is higher than the fluorescence peak. Consequently, the calculated FLH has a negative value, which is the reason for the large number of negative values in the GOCI-derived FLH product. Therefore, it may be inappropriate to select the 680 nm band of GOCI for calculating the fluorescence peak of the FLH.

### 3.2. Fluorescence Signal for Different Water Bodies

Figure 4 shows the spectrum of Rrs under different water conditions (see Table 2 for component concentrations). As a comparison, we changed the concentration of the corresponding ocean color elements to observe its effect on the fluorescence signal. It can be seen that, in different water types, the change of CDOM has basically a negligible effect on the fluorescence peak wavelength (Figure 4a). However, when the concentration of TSM is high, the near-infrared band is affected by the scattering of TSM, obscuring the fluorescence peak (Figure 4b). In general, in different water types, the concentrations of CDOM and suspended solids (under low-to-moderate turbidity) have a limited effect on the migration of fluorescence peaks, and the fluorescence peak wavelengths are closer to 685 than 680 nm. These results are consistent with the findings of Xing [28].

In addition, we also analyzed the effect of the SZA on fluorescence peak wavelengths in clear ocean water and continental shelf water with different Chla concentrations (Figure 5). It can be seen that the chlorophyll fluorescence peak is closer to 685 than to 680 nm, whether for clear ocean water or continental shelf water. The simulation results indicate that, because the value of the 680 nm band is too small, and the Rrs on the baseline is higher than that of the 680 nm band, negative FLH values are produced (as shown in Figure 1). In addition, the simulation results in Figure 5 also reveal that the fluorescence peak of FLH is not affected by the SZA, whereas the inversion model accuracy of chlorophyll is usually greatly affected by the SZA [29,30]. Therefore, establishing an FLH algorithm suitable for

GOCI and applying the FLH algorithm suitable for GOCI to the processing of GOCI data will have more advantages in the study of diurnal variation of ocean water bodies.

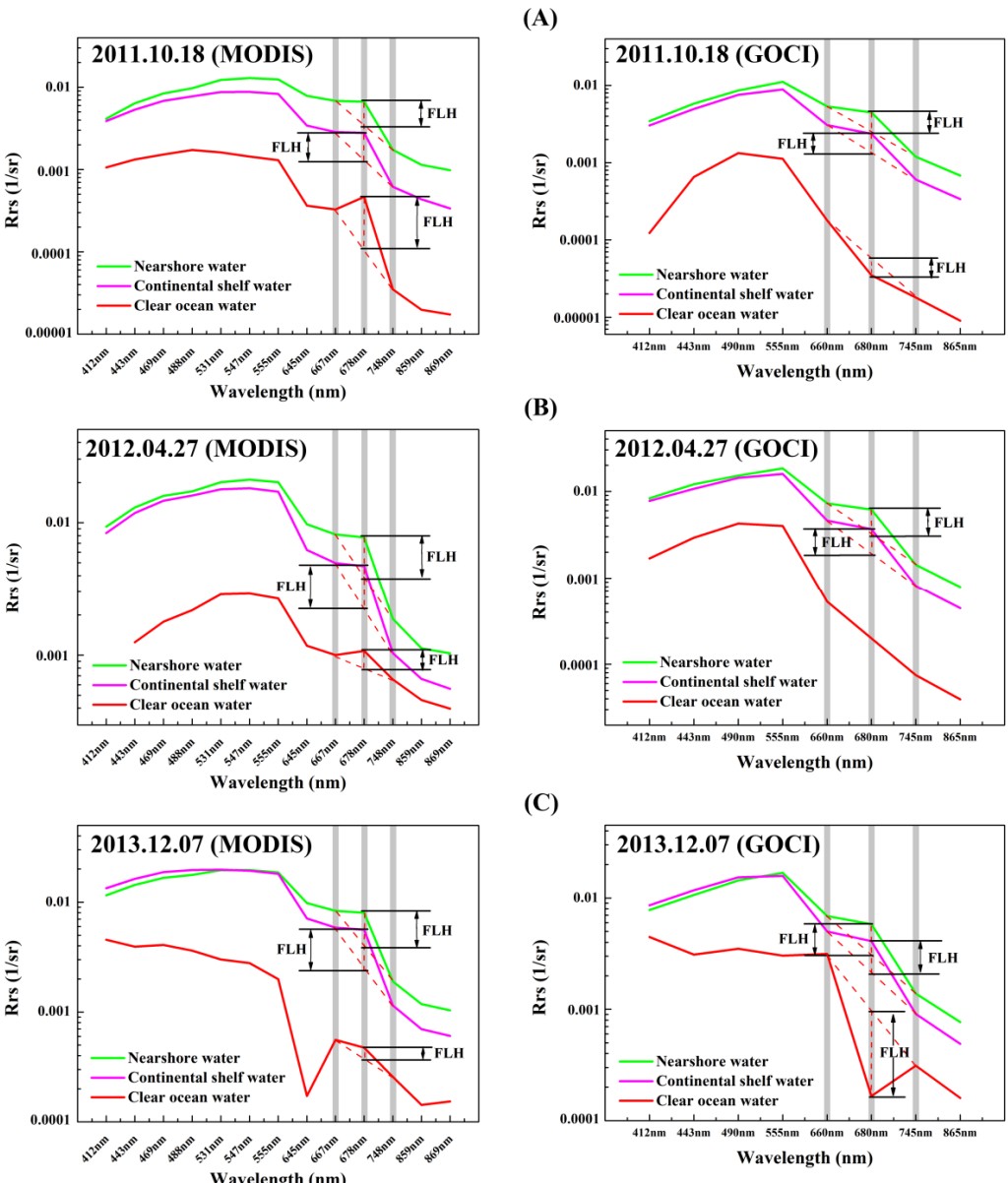

**Figure 3.** Variation in the average Rrs values of different water bodies in different bands for observation dates of (**A**) 18 October 2011, (**B**) 27 April 2012, and (**C**) 7 December 2013. The left sides show the MODIS curves, where the gray histograms denote the three fluorescence bands of MODIS (667, 678, and 748 nm); the right sides show the GOCI curves, where the gray histograms denote the three fluorescence bands of GOCI (660, 680, and 745 nm).

### 3.3. Band Conversion Method

To correct the underestimation of the GOCI FLH product, we used the HydroLight simulated spectral dataset to establish a conversion formula between the Rrs values of 680 and 685 nm, and the results are shown in Figure 6. In terms of band conversion, we used 6048 sets of HydroLight simulation results from seven bands (660, 667, 678, 680, 685, 745, and 748 nm) and performed linear fitting between each band, obtaining the fitting formula as the band conversion equation. The GOCI FLH product can be calculated by using the converted bands and substituting them into Equation (2). In addition, to enable the GOCI FLH results to be compared with the MODIS FLH products, the solar irradiance (F0) of the

mean sun–earth distance was multiplied by Rrs to get the FLH in the form of normalized water-leaving radiance.

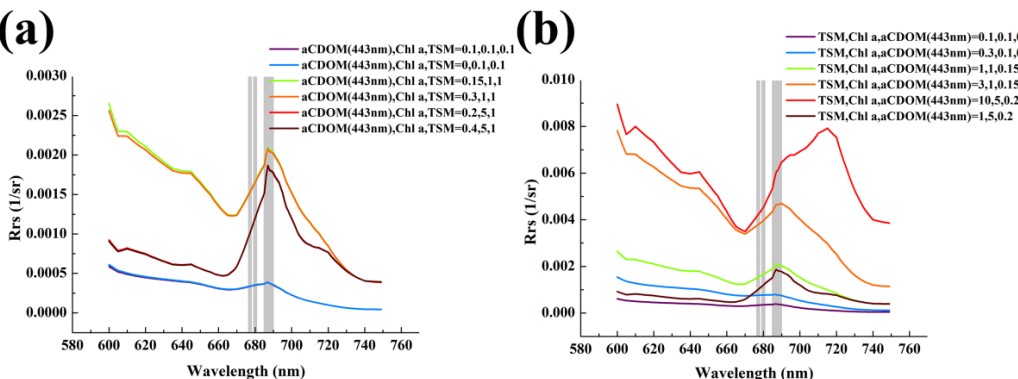

**Figure 4.** Effects of (**a**) aCDOM (443 nm) and (**b**) TSM changes in different water bodies. The three gray histograms denote the positions of the 678, 680, and 685–690 nm bands, respectively.

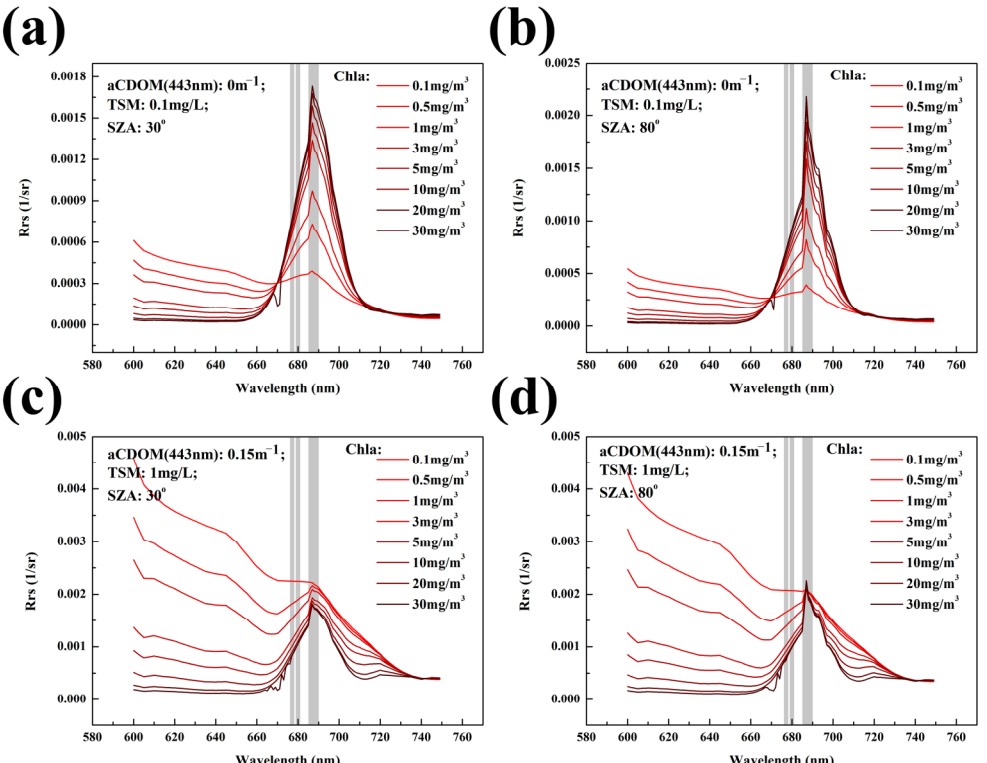

**Figure 5.** Effects of different zenith angles and different Chla concentrations in (**a**,**b**) clear ocean water bodies and (**c**,**d**) continental shelf water bodies. The three gray histograms denote the positions of the 678, 680, and 685–690 nm bands, respectively.

Multi-source satellite data fusion can enhance the spatial and temporal resolution of data and improve data utilization. To improve the accuracy of the MODIS algorithm and then compare and fuse it with the GOCI FLH results, we also converted the three bands selected by MODIS in the same way. The bands used by MODIS FLH are 667, 678, and 748 nm. Based on the simulated dataset, linear fitting was used to convert these bands to 660, 685, and 745 nm, respectively. The MODIS FLH results were recalculated using the band-converted Rrs and compared with the improved GOCI FLH results obtained in the same region at the same time.

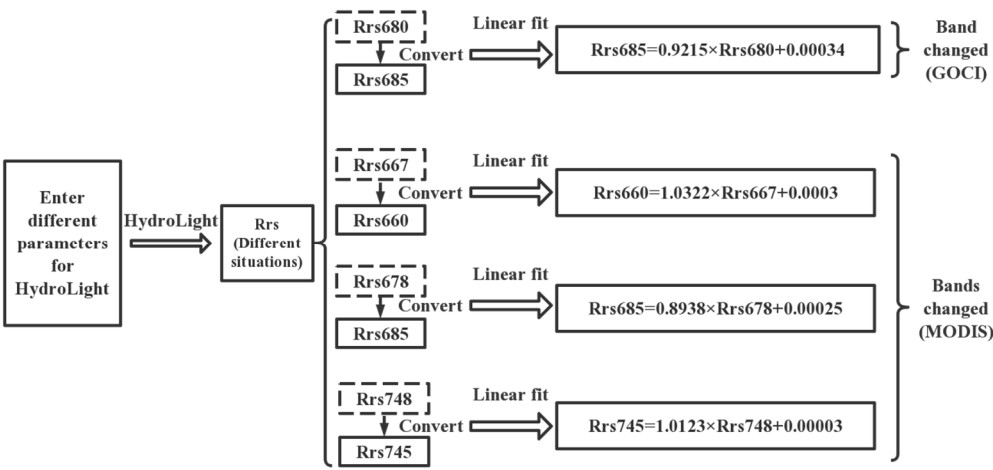

**Figure 6.** Band conversion flowchart.

## 4. Results and Discussion

### 4.1. Algorithm Verification

By using the measured data from two cruises in the East China Sea, the accuracy of the FLH original band algorithm and the improved band algorithms for GOCI and MODIS were tested (Figure 7 and Table 3). When verifying the accuracy of the improved band algorithms for GOCI, we used the in situ Chla to distinguish different type waters. Owing to the limited amount of measured data, we divided them into clear ocean water and continental shelf water with Chla < 5 mg/m$^3$ and nearshore water with Chla > 5 mg/m$^3$ (Figure 7a). When Chla < 5 mg/m$^3$, the FLH value calculated using the original GOCI's 680 nm band is obviously underestimated, and there are many negative values ($R^2$ = 0.61, RMSD = 0.32 sr$^{-1}$, APD = 105.72%, and RPD = −35.02%, respectively). The negative value of the FLH algorithm established in this study is significantly reduced, and the results are basically around the 1:1 line. The overall correlation coefficient is 0.67, and RMSD = 0.16 sr$^{-1}$, APD = 41.92%, and RPD = 19.09%. When Chla > 5 mg/m$^3$, the results shown in Figure 7a are not very good, but the improved band algorithms for GOCI are still better than the original GOCI's 680 nm band. In addition, the statistical parameters obtained by using all the measured data to test the accuracy of the algorithm are $R^2$ = 0.73, RMSD = 0.28 sr$^{-1}$, APD = 35.46%, and RPD = 6.95%. As a polar-orbiting satellite, Aqua only observes once a day in low-to-moderate latitude areas, and the number of matchups is much less than that of GOCI, and the final number of matching points is 17 (Figure 7b). Using the algorithm after band conversion also slightly improves the accuracy of the MODIS FLH product. It should be noted that, in the complex coastal ocean, in-water TSM and CDOM both affect the detection of chlorophyll fluorescence signals. We also simulated the effects of different concentrations of TSM and different absorption coefficients of CDOM (443 nm) (Figure A1 showed in Appendix A) and found that the fluorescence peak would be weakened or disappeared, which masked by the high signal of CDOM and TSM. Thus, the performance of our algorithm in the high-concentration FLH region would be underestimated to some extent, or even fail when fluorescence peak totally disappeared. However, in general, the accuracy of the improved FLH algorithm is much higher than that of the GOCI original algorithm.

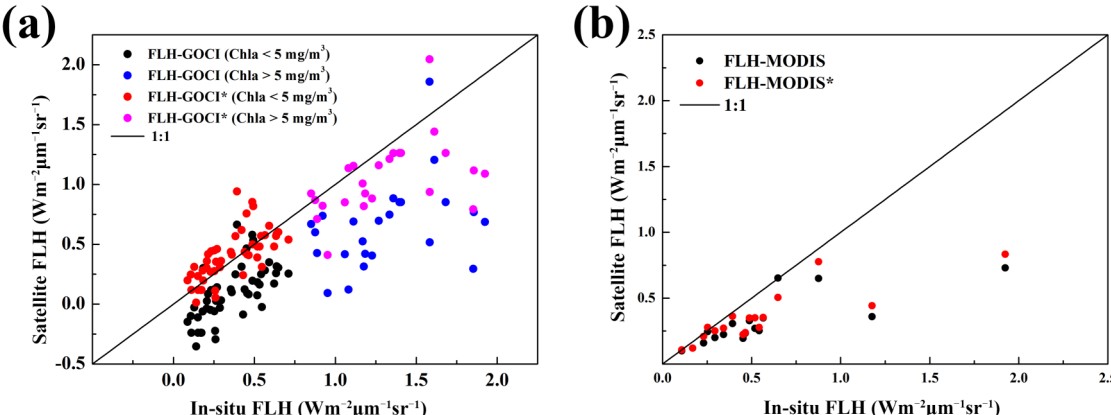

**Figure 7.** Scatter plots of measured data and calculated FLH results from (**a**) GOCI and (**b**) MODIS (FLH-GOCI: FLH result calculated by using the GOCI original band; FLH-GOCI*: FLH algorithm result of GOCI in this study; FLH-MODIS: FLH value calculated by using the MODIS original band; FLH-MODIS*: FLH calculation result of MODIS in this study).

**Table 3.** Results of FLH algorithm performance validated using the measured datasets.

| Algorithm | N | $R^2$ | RMSD $(sr^{-1})$ | APD (%) | RPD (%) |
|---|---|---|---|---|---|
| FLH-GOCI (Chla < 5 mg/m$^3$) | 52 | 0.61 | 0.32 | 105.72 | −35.02 |
| FLH-GOCI* (Chla < 5 mg/m$^3$) | 52 | 0.67 | 0.16 | 41.92 | 19.09 |
| FLH-GOCI (Chla > 5 mg/m$^3$) | 25 | 0.14 | 0.77 | 51.60 | −50.21 |
| FLH-GOCI* (Chla > 5 mg/m$^3$) | 25 | 0.20 | 0.43 | 22.02 | −18.29 |
| FLH-MODIS | 17 | 0.63 | 0.39 | 35.10 | −35.02 |
| FLH-MODIS* | 17 | 0.74 | 0.35 | 28.85 | −27.75 |

When matching with the satellite FLH products, only 30 sets of data remain in the final match due to cloud mask. The results are shown in Figure 8 and listed in Table 4. It is worth mentioning that these included 15 negative FLH values calculated by using the GOCI's original 680 nm band, which could not be inverted into Chla values; the remaining 15 points of Chla values are obviously underestimated ($R^2$ = 0.29, RMSD = 0.30 sr$^{-1}$, APD = 65.60%, and RPD = −63.83%, respectively). However, when we used the FLH algorithm established in this study, the negative values are significantly reduced, and the results are basically around the 1:1 line (with $R^2$ = 0.54, RMSD = 0.13 sr$^{-1}$, APD = 33.10%, and RPD = 11.40%), which demonstrates that the FLH algorithm in this study performs well in all seasons.

### 4.2. Cross-Comparison of GOCI and MODIS FLH Products

The accuracy verifications with in situ data demonstrate the reliability of the algorithm established in this study. Therefore, the new algorithm can be applied to the GOCI and MODIS satellite data (Figure 9). It can be seen that, by using the improved FLH algorithm, the GOCI FLH product has yielded more effective values in different regions and different seasons. Compared with Figure 1, Figure 9 shows that the problem of negative values has been significantly resolved. Compared with the FLH product of MODIS, the GOCI FLH product has a similar magnitude range and basically similar distribution patterns (Figure 10).

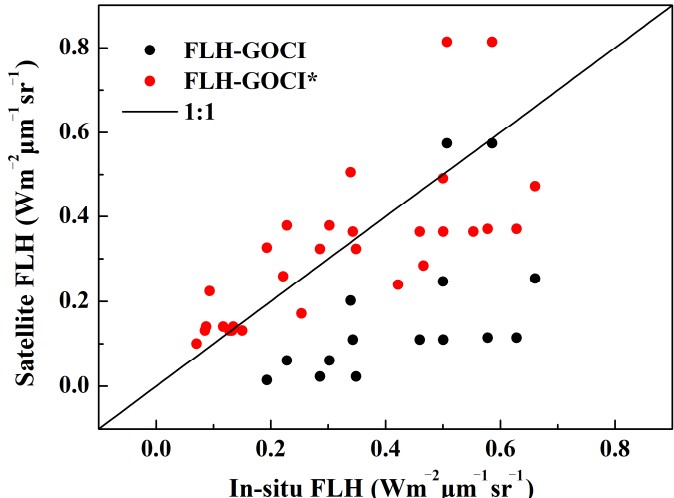

**Figure 8.** Scatter plot of measured data and calculated FLH results (FLH-GOCI: FLH result calculated by using the GOCI original band; FLH-GOCI*: FLH algorithm result of GOCI in this study).

**Table 4.** Results of FLH algorithm performance validated using the measured datasets.

| Algorithm | $N$ | $R^2$ | RMSD ($sr^{-1}$) | APD (%) | RPD (%) |
|-----------|-----|-------|------------------|---------|---------|
| FLH-GOCI | 15 | 0.29 | 0.30 | 65.60 | −63.83 |
| FLH-GOCI* | 30 | 0.54 | 0.13 | 33.10 | 11.40 |

Multiple factors have caused the difference in the FLH products of the two satellites: 1. The spatial resolution of the two satellites is different, with that of GOCI being 1 km and that of MODIS being 500 m. Although the resampling method was used, a certain amount of error remains. 2. The signal-to-noise ratios of the satellites are different (e.g., that of the 683 nm band of MODIS is 1087, while that of the 680 nm band of GOCI is 870). 3. The difference between different satellite products is reduced by band conversion, but it cannot be completely eliminated. However, in general, the distribution of FLH products of the two satellites are consistent (Figure 9), and the scattering points are basically around the 1:1 line (Figure 10), demonstrating that the effect of band conversion is relatively small. After band conversion, the baseline band value used for the MODIS FLH calculation is lower than that of GOCI, resulting in the MODIS FLH being slightly greater than that of GOCI.

*4.3. Application of Improved FLH Algorithm to GOCI Hourly Data*

To further evaluate the diurnal stability of the established FLH algorithm, the algorithm was applied to the hourly data from GOCI. Figure 11A shows the hourly FLH results calculated using the original GOCI band on 18 May 2017. It can be seen that, in the nearshore region, owing to the high fluorescence peak, a positive value can be obtained by inversion, whereas, in offshore waters, there are many negative values (as shown in the blank area in the figure). Figure 11B shows the inversion results of the GOCI data obtained by using the improved FLH algorithm. It can be seen that the inversion results of the improved algorithm have sufficient effective values, and the results show the diurnal stability of the performance of the improved algorithm. The statistical results of the data efficiency are provided in Table 5, and it can be seen that the effective FLH data from the original band can only reach an average coverage rate of 14.78%, but the improved algorithm can achieve an average coverage rate of 66.73%. Therefore, compared with the FLH calculated by using the GOCI original band, the improved algorithm can effectively solve the problem of negative FLH values.

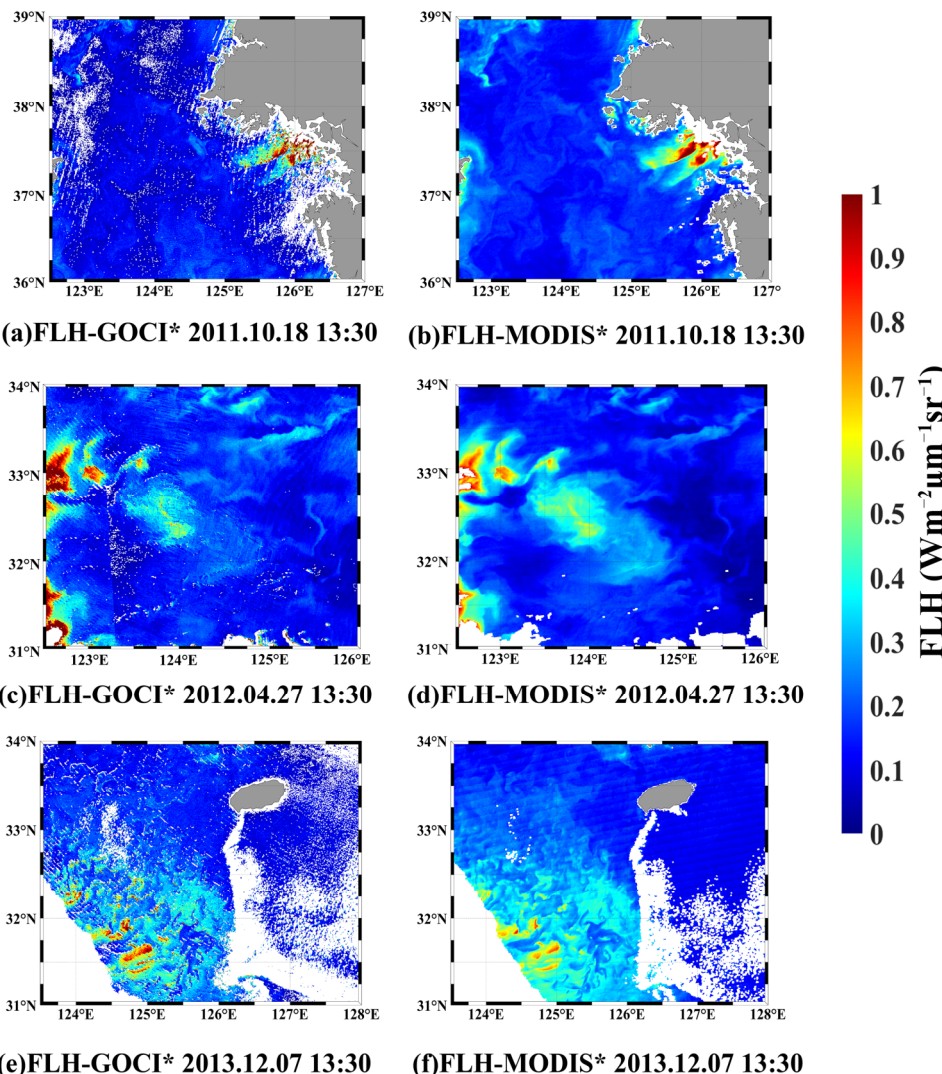

**Figure 9.** Comparison of improved GOCI FLH results with improved MODIS FLH results.

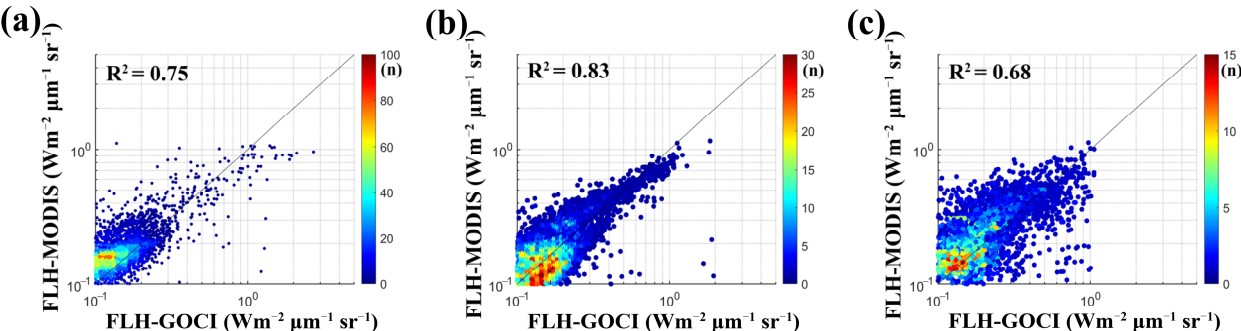

**Figure 10.** Density scatter plots between GOCI and MODIS FLH products using the improved algorithms. (**a**) 18 October 2011, (**b**) 27 April 2012, and (**c**) 7 December 2013.

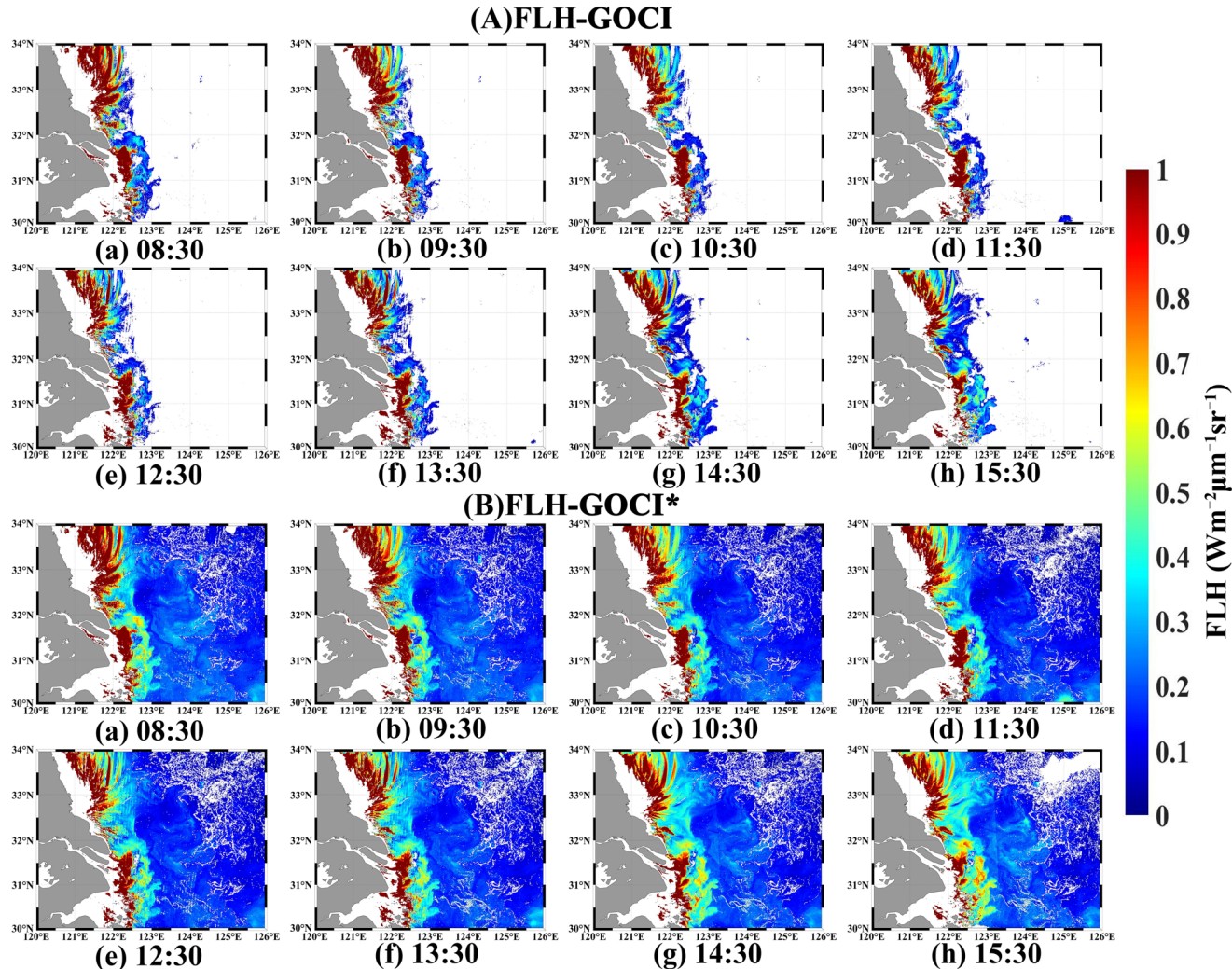

**Figure 11.** Comparison of FLH products generated using the pre-improved (FLH-GOCI) and post-improved (FLH-GOCI*) algorithms for 8 images per day from GOCI (18 May 2017).

**Table 5.** Effective data coverage (in percent) before and after the improvement of the FLH algorithm of GOCI. (The ratio of the effective value to the number of all pixels in the area is taken as the coverage of the effective value.)

| Time | 8:30 | 9:30 | 10:30 | 11:30 | 12:30 | 13:30 | 14:30 | 15:30 | 8-Scene Average |
|---|---|---|---|---|---|---|---|---|---|
| FLH-GOCI* | 65.87 | 67.22 | 67.03 | 65.65 | 67.00 | 68.35 | 68.12 | 64.60 | 66.73 |
| FLH-GOCI | 14.27 | 13.34 | 12.18 | 12.47 | 13.84 | 15.53 | 17.69 | 18.93 | 14.78 |

Based on the hourly GOCI FLH obtained by the improved algorithm on 18 May 2017, we selected three typical areas and analyzed the diurnal variation of FLH (Figure 11). It can be seen that, because of the high dynamic amounts of TSM in the coastal waters, the diurnal fluctuation of FLH is also significant. In contrast, in continental shelf water and clear ocean water, because of the lesser influence by tidal disturbances and terrestrial inputs, the diurnal variations of FLH are less affected from morning to night. Interestingly, it can be seen from Figure 12b that the FLH value around 11:30 is low, which might be linked to the vertical migration of phytoplankton, moving to the deep layer during the strong light incident at noon [31].

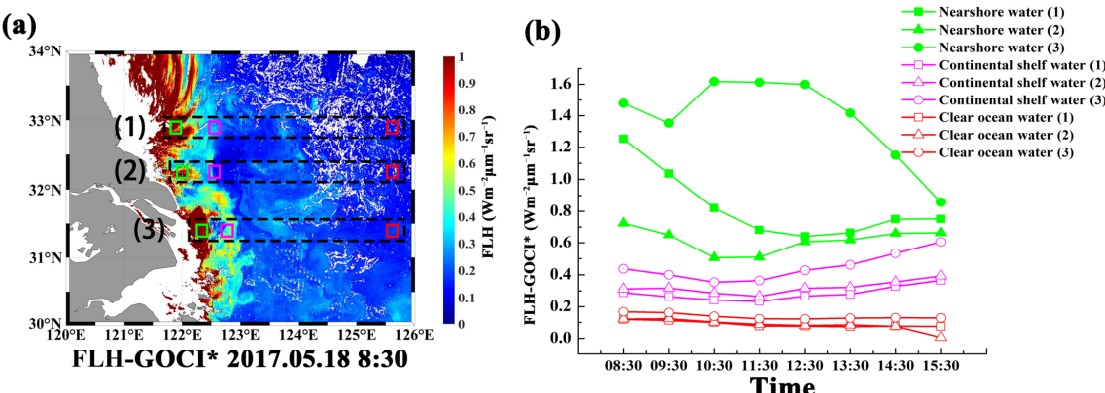

**Figure 12.** Hourly FLH changes of three water bodies on 18 May 2017. (The framed area (1–3) in panel (**a**) is the selected area, the green frame is the nearshore water area, the purple frame is the continental shelf water area, and the red frame is the clear ocean water area.) and (**b**) is the hourly FLH changes of three water bodies on 18 May 2017.

Because MODIS and GOCI have different spatial resolutions, we first performed a gridded projection onto the same area with the spatial resolution of 1 km to fuse the data. Specifically, a location matrix is established, and then the pixel values of the nearest neighbors of different satellites are assigned to the new pixels. If the pixel point has only one satellite data, the data is used. If the pixel point has two satellite data, we take the average value. Finally, we output the combined result, including latitude and longitude and pixel-by-pixel values. Figure 13 shows the combined results of the improved GOCI and MODIS satellite FLH products on 27 April 2012. In different regions, the effective data coverage of the fusion product for different water bodies has been improved, and the noise problem in GOCI FLH has also been significantly resolved. Through the band conversion method proposed in this study, data from different satellites can be better integrated, and the spatial and temporal resolution of the data can be improved.

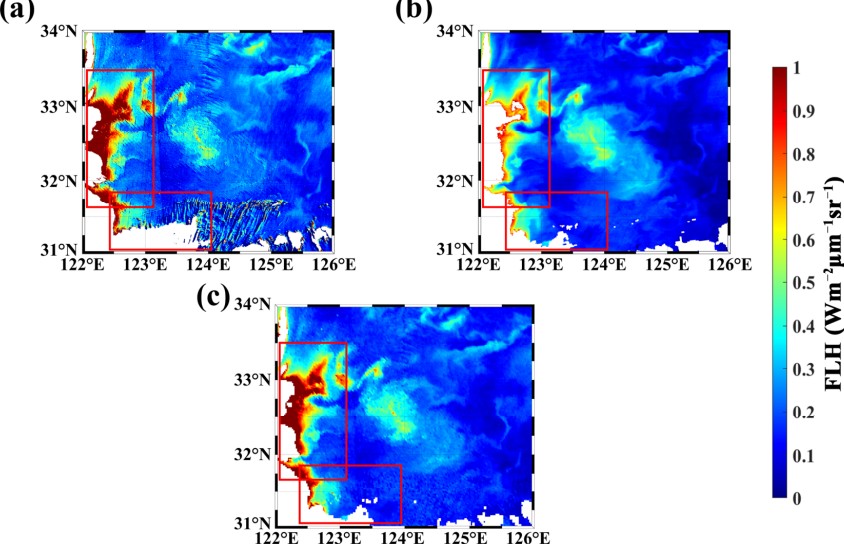

**Figure 13.** Satellite FLH product on 27 April 2012: (**a**) GOCI satellite FLH product (calculated using the new algorithm in this study), (**b**) MODIS satellite FLH product (calculated using the new algorithm in this study), and (**c**) fusion FLH product results. The red frames are the regions where the fusion results are more significant.

## 5. Conclusions

In this study, we proposed a new FLH algorithm for GOCI. The new algorithm converts the fluorescence peak band from 680 to 685 nm according to the water spectrum dataset simulated using HydroLight. After band conversion, the new FLH algorithm can effectively resolve the problem of the numerous negative values generated by using the original band. We used the in situ data obtained from two cruises in the East China Sea to evaluate the accuracy of the GOCI FLH algorithm established in this study, and the results showed that the new FLH algorithm has high accuracy with a correlation coefficient of 0.7302, RMSD of 0.27577 $sr^{-1}$, APD of 35.46%, and RPD of 6.95%. The new FLH algorithm can be applied to hourly GOCI data to monitor the diurnal variation of phytoplankton fluorescence line height, thereby providing more information on phytoplankton biomass and physiological changes. In addition, the FLH satellite products generated by the band conversion algorithm can be used to merge FLH products from multiple ocean color satellite missions to improve data coverage and extend the time series span.

**Author Contributions:** Conceptualization, M.Z. and Y.B.; methodology, M.Z.; software, H.L.; validation, X.H., F.G. and T.L.; formal analysis, M.Z.; investigation, M.Z. and Y.B. All authors have read and agreed to the published version of the manuscript.

**Funding:** This study was supported by the Key Special Project for Introduced Talents Team of the Southern Marine Science and Engineering Guangdong Laboratory (Guangzhou) (GML2019ZD0602), the National Natural Science Foundation of China (Grants Nos. 42176177, 41825014 and 42141002), and the Zhejiang Provincial Natural Science Foundation of China (2017R52001 and LR18D060001).

**Data Availability Statement:** Not applicable.

**Acknowledgments:** We thank the Korea Institute of Ocean Science & Technology/Korea Ocean Satellite Center for providing the GOCI data. We thank the National Aeronautics and Space Administration for providing the MODIS data. We thank Young-Je Park, Hak-Yeol You, Jae-Seol Shim, Joo-Hyung Ryu, and their staff for collecting the Chla data at two AERONET-OC sites (Gageocho_Station and Ieodo_Station).

**Conflicts of Interest:** The authors declare no conflict of interest.

## Appendix A

In the complex coastal ocean, in-water TSM and CDOM both affect the detection of chlorophyll fluorescence signals. We also simulated the effects of different concentrations of TSM and different absorption coefficients of CDOM (443 nm) (Figure A1), and found that the fluorescence peak would disappear, which masked by the high signal of CDOM and TSM, so the inversed results in the high-concentration FLH region would be underestimated to some extent.

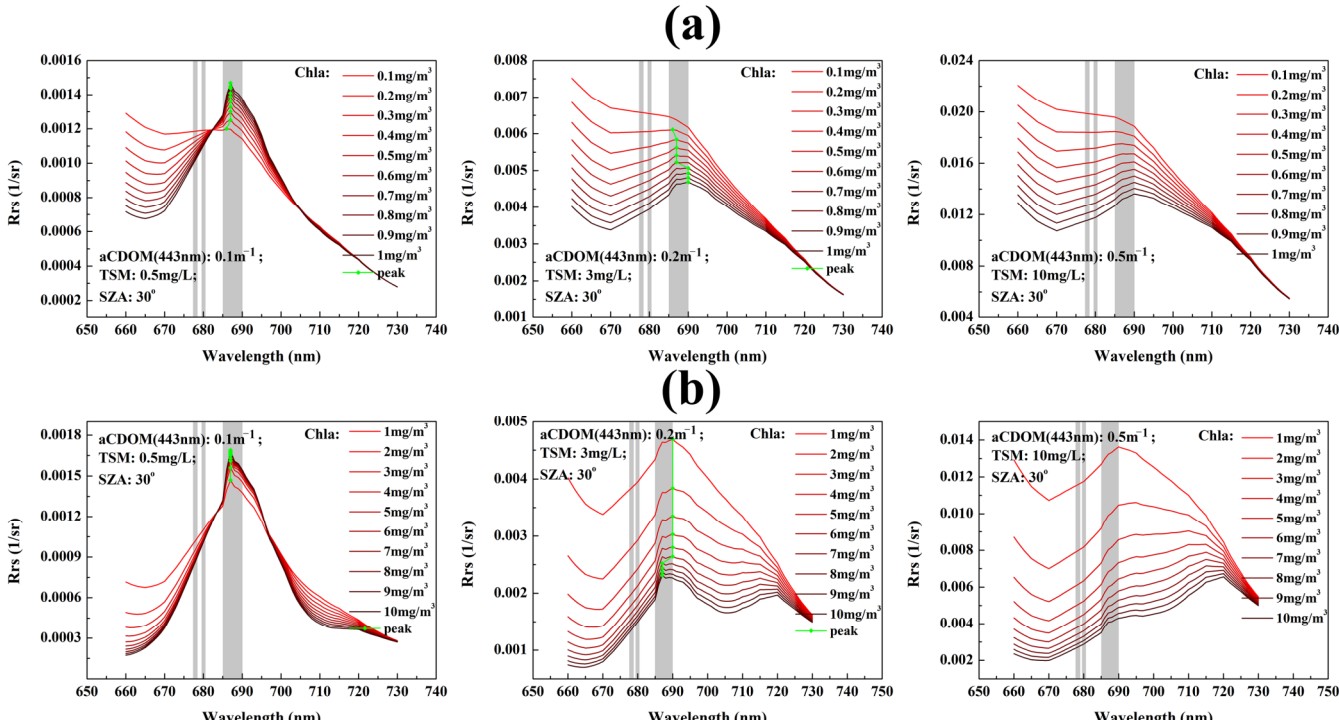

**Figure A1.** Effects of CDOM and TSM changes in different Chla water bodies. (**a**) Chla are 0.1–1 mg/m³, and (**b**) Chla are 1–10 mg/m³.

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
