# Peer review of "Fluorescence Line Height Extraction Algorithm for the Geostationary Ocean Color Imager"

_remotesensing, doi:10.3390/rs14112511_

Round 1
Reviewer 1 Report
This paper propose a method in order to get assessments and improve the existing ones for fluorescence line height (FLH) provided by GOCI sensors. The paper highlights why the GOCI algorithms failed at retrieving FLH on many areas: concern with the GOCI 685nm spectral band. The proposed method is to simulate a spectral band that is not in GOCI products (685nm band simulated from 680nm band) in order to use the simulation and assess FLH thanks to a classical method applied on the resulting spectral band. The proposed algorithm is applied on both GOCI and MODIS data. Results are compared with 77 in situ measurements. There are some minor concerns listed hereafter.
Figures: text size is too small.
Resolution of MODIS (1km) and GOCI (500m) are different. When using the method exposed in §2.5 based on a 3 x 3 pixel spatial window, a larger area is used with MODIS when compared with the area used for GOCI. How could this affect the comparisons? And was this handle in the fusion process (§4.3)?
Fig 1: is it possible to show framed areas on (a), (c) and (e) too?
§2.1: it is probably better to use a “normalized time description”: 8:30am to 3:30pm -> 8:00 to 15:00 (l139) or 13:30 -> 1:30pm (l152)
Fig 3: Does it mean that there is an inappropriate calibration of GOCI sensors?
§3.2 (Fig.5): “When the Chla concentration is low (<0.5 mg/m3), the chlorophyll fluorescence peak is close to 683 nm. When Chla concentration is 0.5–5 mg/m3, the chlorophyll fluorescence peak is close to 685 nm. When the Chla concentration is >5 mg/m3, the chlorophyll fluorescence peak is closer to 690 nm. Therefore, it can be shown that the chlorophyll fluorescence peak is closer to 685 than to 680 nm, whether for clear ocean water or continental shelf water.”
I do not see the peak sliding according to chla concentration on Fig. 5. I never see the peak on 690 nm. I may miss something? However, this figure clearly shows “that the chlorophyll fluorescence peak is closer to 685 than to 680 nm”.
§3.3: Could you briefly explain how was set the method applied for the band conversion? In addition, this linear relation means that whatever the conditions, we can apply the same conversion between two spectral bands. Is there any reference about this result?
§4.1 (l329): writing “R²=0.6117, RMSD = 0.51302 sr-1, …” (like on line 332) would make the reading easier.
§4.2 (Fig.9): It seems there are some patterns on the density scatter plots, especially seen on (B) and (c). How do you interpret and explain them?
§4.3 (l396): How was processed the fusion? You could explain concerns about resolution and other aspects (pixel gridding, etc.).
Author Response
Response to Reviewer 1 Comments
Point 1: Figures: text size is too small.
Response 1: Thanks for this comment. We have adjusted the font size of the text in all Figures.
Point 2: Resolution of MODIS (1km) and GOCI (500m) are different. When using the method exposed in §2.5 based on a 3 x 3 pixel spatial window, a larger area is used with MODIS when compared with the area used for GOCI. How could this affect the comparisons? And was this handle in the fusion process (§4.3)?
Response 2: Thanks for this comment. The spatial resolutions of MODIS and GOCI are indeed different, so this has some impact on data fusion and comparison with field data.
For the matching of measured data, we followed the validation method proposed by Bailey and Werdell (2006). The satellite match-up dataset comprised the mean of the valid values found for the 3x3 pixel within a time window of 3 hours from the in situ observation. It is true that a 3x3 pixel box means a large area for MODIS, which may result in slightly poor evaluation results, but using a multi-pixel box has the following advantages: 1) The use of a multi-pixel box could result in more available mathup data with the in situ data. 2). The spatial consistency of the pixel frame is evaluated through the criteria in Section 2.5, so the average value of the pixels in the cell box can represent the value of the center pixel.
For different spatial resolutions, we compare the averaged results of GOCI by 3x3 pixel and 7x7 pixel. Among them, the 3x3 pixel frame is the one used in this study, and the GOCI space size of the 7x7 pixel frame (3500m) is close to that of MODIS (3000m). It can be seen that the correlation coefficient is 0.99 (Fig S1), and the relative deviation is small, so the size of the pixel frame brings less affected.
Fig. S1. Comparing the GOCI FLH results of 3x3 pixel and 7x7 pixel averaged. (FLH-GOCI*: FLH algorithm result of GOCI in this study.)
For the second question, due to the different spatial resolutions of MODIS and GOCI, we first performed gridded projection onto the same map of area at a spatial resolution of 1 km. Then perform the fusion. The effect of spatial resolution is eliminated by projecting onto the same spatial resolution.
Point 3: Fig 1: is it possible to show framed areas on (a), (c) and (e) too?
Response 3: Thanks for this suggestion. We have added framed areas on Fig. 1 (a), (c) and (e).
Point 4:§2.1: it is probably better to use a “normalized time description”: 8:30am to 3:30pm -> 8:00 to 15:00 (l139) or 13:30 -> 1:30pm (l152).
Response 4: Thanks for this suggestion. We have change the time description to “8:00 to 15:00” in the revision.
Point 5: Fig 3: Does it mean that there is an inappropriate calibration of GOCI sensors?
Response 5: Thanks for this comment. In Fig. 3, We did not mean there is an inappropriate calibration of GOCI sensors, this picture is to show that the corresponding Rrs value at 680nm is too low, so that the calculated FLH value is mostly negative. Ryu et al. (2012) tested the accuracy of GOCI's Rrs products based on the measured data around the Korean peninsula, and reported that the relative deviation in the 680nm band is 28.3%, which is much larger than 18.3% in the 550nm band. GOCI sensor parameters reported by Ryu et al. (2012), showing in Table s1, the signal-to-noise ratio of the 680 band is also much lower than that of the visible light band.
Table s1. GOCI sensor parameters. Ryu et al. (2012)
Band |
Band center (nm) |
Band width (nm) |
Nominal radiance [mW/(cm2*μm*sr)] |
Signal-to-noise ratio (SNR) |
B1 |
412 |
20 |
10.0 |
1000 |
B2 |
443 |
20 |
9.25 |
1090 |
B3 |
490 |
20 |
7.22 |
1170 |
B4 |
555 |
20 |
5.53 |
1070 |
B5 |
660 |
20 |
3.20 |
1010 |
B6 |
680 |
10 |
2.71 |
870 |
B7 |
745 |
20 |
1.77 |
860 |
B8 |
865 |
40 |
1.20 |
750 |
Point 6:§3.2 (Fig.5): “When the Chla concentration is low (<0.5 mg/m3), the chlorophyll fluorescence peak is close to 683 nm. When Chla concentration is 0.5–5 mg/m3, the chlorophyll fluorescence peak is close to 685 nm. When the Chla concentration is >5 mg/m3, the chlorophyll fluorescence peak is closer to 690 nm. Therefore, it can be shown that the chlorophyll fluorescence peak is closer to 685 than to 680 nm, whether for clear ocean water or continental shelf water.”
I do not see the peak sliding according to chla concentration on Fig. 5. I never see the peak on 690 nm. I may miss something? However, this figure clearly shows “that the chlorophyll fluorescence peak is closer to 685 than to 680 nm”.
Response 6: Thanks for this comment. We have conducted a lot of simulations. Due to the space limitations in the main text, only a part of the results is shown in Fig. 6 in revised manuscript. As our purpose was to prove that the Chla peak is closer to 685nm than 680nm, we have deleted the other sentences about peak sliding to avoid the misleading.
Point 7:§3.3: Could you briefly explain how was set the method applied for the band conversion? In addition, this linear relation means that whatever the conditions, we can apply the same conversion between two spectral bands. Is there any reference about this result?
Response 7: For the first question, the linear conversion is mainly aimed at the adjacent bands with small wavelength difference. Since the optical properties of the water body and the atmosphere are similar, there is a good linear relationship between the Rrs. In this paper, based on 6048 simulation data, we found there are good linear relationships between bands, with the correlation coefficient of 0.99. Therefore, the linear relationship can describe the conversion relationship between two bands. Since the simulation data covers different water body types, this linear relationship has good wide applicability. Of course, there are differences in the coefficients of the linear relationship between different bands, and the linear model needs to be rebuilt when applying to new bands.
For the second question, there were many studies about band shifting for ocean color multi-spectral reflectance data (Valiente et al. 1995; Gordon et al. 1995; Mélin, et al. 2015;). Mélin, et al. (2015) proposed an approach to perform band shifting applied to multi-spectral ocean remote sensing reflectance Rrs values in the visible spectral range. Based on the determination of inherent optical properties (IOPs) and a bio-optical model, this band-shifting approach expressed Rrs at neighboring bands. The results show that in the visible light band, the relative error is basically within 5%. In the chlorophyll-sensitive band (such as 490nm), the relative error is within 1%. Fig. S2 shows the results of applying band conversion to different satellites. It can be seen that they have good performance.
Fig. S2. Average of all coincident spectra for the SeaWiFS data (red diamonds), the original MODIS data (grey line and crosses) and MODIS band-shifted data (black circles) (the Fig S2. from Frédéric, et al. 2015)
Point 8:§4.1 (l329): writing “R²=0.6117, RMSD = 0.51302 sr-1, …” (like on line 332) would make the reading easier.
Response 8: Thanks for this suggestion. We have corrected it in the revision.
Point 9:§4.2 (Fig.9): It seems there are some patterns on the density scatter plots, especially seen on (B) and (c). How do you interpret and explain them?
Response 9: Thanks for this comment. Grading is required when drawing a density scatter plot. The original grading is too small (200 grades), which leads to the phenomenon of plaques. In the revised manuscript, we use a grading number of 1000, so this effect can be better eliminated.
Point 10:§4.3 (l396): How was processed the fusion? You could explain concerns about resolution and other aspects (pixel gridding, etc.).
Response10: Thanks for this comment. As the reply to the question above, because MODIS and GOCI have different spatial resolutions, we first performed a gridded projection onto the same area with the spatial resolution of 1 km to fuse the data. Specifically, a location matrix is established, and then the pixel values of the nearest neighbors of different satellites are assigned to the new pixels. If the pixel point has only one satellite data, the data is used. If the pixel point has two satellite data, we take the average value. Finally, we output the combined result, including latitude and longitude and pixel-by-pixel values. We have added this description to the manuscript.
References:
- Bailey, S.W.; Werdell, P. A multi-sensor approach for the on-orbit validation of ocean color satellite data products. Remote Sens. Environ. 2006, 102(1-2): 12-23.
- Ryu, J.H.; Han, H.J.; Cho, S.; et al. Overview of geostationary ocean color imager (GOCI) and GOCI data processing system (GDPS). Ocean S J. 2012, 47(3): 223-233.
- Valiente, J.A., Nunez, M., Lopez-Baeza, E., et al. Narrow-band to broad-band conversion for Meteosat-visiible channel and broad-band albedo using both AVHRR-1 and-2 channels. Remote Sens. 1995, 16(6): 1147-1166.
- Gordon, H.; Remote sensing of ocean color: a methodology for dealing with broad spectral bands and significant out-of-band response. Appl. Optics 1995, 34(36): 8363-8374.
- Mélin, F.; Sclep, Band shifting for ocean color multi-spectral reflectance data. Opt. Express 2015, 23(3): 2262-2279.

Reviewer 2 Report
Review report on “Fluorescence Line Height Extraction Algorithm for the Geostationary Ocean Color Imager” by Min Zhao et al.
- Lines 111-121: The calculation method and formula of Kim [18] need to be explained in more detail here, because this is the starting point of the story of this manuscript, which is very important.
- Lines 122-128: Here, the author needs to explain in detail the analysis area of this study, the time of using the data and anything that can help readers quickly understand the research objectives.
- Figure 1: If the author wants to use the map to display the research area, this image must be reorganized. The author needs to give a large area and combined with the ocean topography data, and these three frames can be drawn directly on the map. Then place the existing six subgraphs close to it.
- Figure 2: what is “data @ Dummy=first” mean?
- Table 1: The meaning of these "values" needs to be explained in detail.
- Figure 3: Improve the image resolution and replace "1e-3" with "0.001", or use log coordinates.
- Figure 4: Improve the image resolution.
- Figure 7 & Table 3: It seems that the results of high concentration FLH corresponding to GOCI and MODIS data are not good. Obviously, the value calculated by the satellite is much lower than the in-situ observation value. Why? The author should distinguish the sample points according to the three areas mentioned in Table 2 and calculate the corresponding statistical results. Therefore, Figure 7 should have 6 subgraphs, and table 3 also has 6 rows.
- Lines 353-355 and Figure 9: Obviously, the FLH obtained from the two dataset is different. What are the correlation coefficients of the three graphs? Why is the FLH from MODIS data higher than GOCI?
- The authors need to seek professional English editors to improve sentence structure.
Reviewer 3 Report
The manuscript focuses on an innovative algorithm for extracting Fluorescence Line Hight (FLH) estimates using data from the GOCI product, which does not natively support FLH.
The topic falls within the scope of the journal and the results are promising to be incorporated into GOCI data processing in the near future to obtain FLH data as well.
The article is well written and structured and I recommend its publication, although further analysis must be carried out to finalize the algorithm and fully support the Author's claim of lines 418-423. I understand that although the algorithm was tuned using the simulated radiative transfer provided by the HydroLight model compared to the few in situ data (77 stations) performed only in August, the results show a good correlation also in other seasons, but I suggest to the Authors discuss more on this aspect.
Minor remark:
line 152: Please, delete a full stop.
Round 2
Reviewer 2 Report
Please add Figure S1 to the appendix of the manuscript.
